# *Acinetobacter baumannii* IC2 and IC5 Isolates with Co-Existing *bla*_OXA-143-like_ and *bla*_OXA-72_ and Exhibiting Strong Biofilm Formation in a Mexican Hospital

**DOI:** 10.3390/microorganisms11092316

**Published:** 2023-09-14

**Authors:** Julia Moreno-Manjón, Santiago Castillo-Ramírez, Keith A. Jolley, Martin C. J. Maiden, Catalina Gayosso-Vázquez, José Luis Fernández-Vázquez, Valeria Mateo-Estrada, Silvia Giono-Cerezo, María Dolores Alcántar-Curiel

**Affiliations:** 1Laboratorio de Infectología, Microbiología e Inmunología Clínica, Unidad de Investigación en Medicina Experimental, Facultad de Medicina, Universidad Nacional Autónoma de México, Ciudad de México 06720, Mexico; juliamorenomanjon@gmail.com (J.M.-M.); catalina_gayosso@yahoo.com.mx (C.G.-V.); joseluis_f@hotmail.com (J.L.F.-V.); 2Laboratorio de Bacteriología Médica, Posgrado en Ciencias Quimicobiológicas, Escuela Nacional de Ciencias Biológicas, Instituto Politécnico Nacional, Ciudad de México 11350, Mexico; 3Programa de Genómica Evolutiva, Centro de Ciencias Genómicas, Universidad Nacional Autónoma de México, Cuernavaca 62209, Mexico; iago@ccg.unam.mx (S.C.-R.); vmateo@lcg.unam.mx (V.M.-E.); 4Department of Biology, University of Oxford, Oxford OX1 3SZ, UK; keith.jolley@biology.ox.ac.uk (K.A.J.); martin.maiden@biology.ox.ac.uk (M.C.J.M.)

**Keywords:** resistome, biofilm formation, antimicrobial resistance, healthcare-associated infections, outbreak, genomic epidemiology, core genome multilocus sequence typing, *Acinetobacter baumannii*

## Abstract

*Acinetobacter baumannii* is an opportunistic pathogen responsible for healthcare-associated infections (HAIs) and outbreaks. Antimicrobial resistance mechanisms and virulence factors allow it to survive and spread in the hospital environment. However, the molecular mechanisms of these traits and their association with international clones are frequently unknown in low- and middle-income countries. Here, we analyze the phenotype and genotype of seventy-six HAIs and outbreak-causing *A. baumannii* isolates from a Mexican hospital over ten years, with special attention to the carbapenem resistome and biofilm formation. The isolates belonged to the global international clone (IC) 2 and the Latin America endemic IC5 and were predominantly extensively drug-resistant (XDR). Oxacillinases were identified as a common source of carbapenem resistance. We noted the presence of the *bla*_OXA-143-like_ family (not previously described in Mexico), the *bla*_OXA-72_ and the *bla*_OXA-398_ found in both ICs. A low prevalence of efflux pump overexpression activity associated with carbapenem resistance was observed. Finally, strong biofilm formation was found, and significant biofilm-related genes were identified, including *bfmRS*, *csuA/BABCDE*, *pgaABCD* and *ompA.* This study provides a comprehensive profile of the carbapenem resistome of *A. baumannii* isolates belonging to the same pulse type, along with their significant biofilm formation capacity. Furthermore, it contributes to a better understanding of their role in the recurrence of infection and the endemicity of these isolates in a Mexican hospital.

## 1. Introduction

The ESKAPE (*Enterococcus faecium*, *Staphylococcus aureus*, *Klebsiella pneumoniae*, *A. baumannii*, *Pseudomonas aeruginosa*, and *Enterobacter* spp.) group is an important set of bacterial pathogens relevant to human health. Among these species, carbapenem-resistant *A. baumannii* is considered one of the World Health Organization’s critical priority pathogens for which new antimicrobials are required [1]. *A. baumannii* is an opportunistic pathogen that mainly infects critically ill patients. This bacterium causes a wide variety of infections, with pneumonia and bacteremia the most relevant, where the mortality ranges from 14% to 82%, depending on the site of infection and antimicrobial resistance [2,3].

An outstanding feature of *A. baumannii* is its complex antimicrobial resistance mechanisms [2]. The first line of antibiotics that are used when dealing with a sensitive *A. baumannii* infection are broad-spectrum cephalosporins, such as ceftazidime and cefepime; β-lactams in combination with a β-lactamase inhibitor, such as sulbactam; and carbapenems, such as imipenem and meropenem, which are used as a last-resort treatment for HAIs. Importantly, the resistance to carbapenems is increasing [4]. In *A. baumannii,* three carbapenem resistance mechanisms have been described, including (1) the production of carbapenem-hydrolyzing enzymes belonging to Ambler Classes B and D, also known as metallo-β-lactamases (MBLs) and oxacillinases (OXAs), respectively; (2) efflux pump alterations; and, less frequently, (3) loss of porins [5,6]. Currently, the association between the loss of porins and antibiotic resistance is controversial since, despite the decrease in the amount of internalized antibiotics caused by the loss of porins, it also implies a decrease in nutrient intake [7,8,9].

The most common resistance mechanism in *A. baumannii* against carbapenems is oxacillinase enzymes, which hydrolyze the β-lactam antimicrobials [6]. A less common group of β-lactamases enzymes are the metallo-β-lactamases [6]. Efflux pumps are intrinsic to bacteria, and *A. baumannii* has six main efflux pump superfamilies that co-exist: the major facilitator superfamily (MFS); the resistance nodulation division family (RND); the small multidrug resistance family (SMR); the multidrug and toxin extrusion family (MATE); the ATP-binding cassette transporters (ABC); and the proteobacterial antimicrobial compound efflux PACE family [10]. The best-studied efflux pumps are the Ade-type, such as AdeABC, which are part of the RND family and have antiport gradient pumps induced by the presence of different antimicrobials [11,12].

Previous studies have shown that the high levels of *A. baumannii* antimicrobial resistance are associated with biofilm formation, making it one of the most relevant virulence factors for survival in harsh environments, including those of high antimicrobial concentration [3,13]. Biofilms are organized as multicellular communities of bacteria that are surrounded by self-produced exopolysaccharide matrices and are responsible for 65–80% of human infections [14]. The presence of several classes of molecules has been associated with the formation of biofilm in clinical isolates of *A. baumannii*, the most conserved being the chaperone-usher pili (Csu) and OmpA, followed by the biofilm-associated protein (Bap) and PER-1 extended-spectrum class A β-lactamase. Other factors that have been implicated in adherence and biofilm formation include the two components system BfmRS, which upregulates the *csu* pili operon (*csuA/BABCDE*) and the *pgaABCD* operon [15].

Typing methods such as pulsed-field gel electrophoresis (PFGE) and multilocus sequence typing (MLST) are frequently used in molecular epidemiology fingerprinting to study the transmission of pathogens [16,17,18,19]. PFGE fingerprinting is a method for clonality investigation based on the determination of bacterial pulse types that allows the detection of bacterial spread in a hospital and the identification of HAI outbreaks [20], while MLST is a sequence-based method used to identify sequence types (STs) and clonal complexes using housekeeping genes to compare isolates from diverse isolation sources [20,21]. However, due to the highly dynamic genome of *A. baumannii*, MLST does not genotype the isolates of this species to a sufficient level of resolution for clinical purposes [22]. Advances in whole genome sequencing (WGS) technology have provided access to very high-resolution genetic information of isolates at a very large scale, i.e., hundreds or thousands of isolates. This has made it possible to not only determine virulence and resistance mechanisms but also molecular epidemiology to understand the transmission of pathogens [23]. Core-genome multilocus sequence typing (cgMLST) is a genotyping method that indexes hundreds to thousands of core genes, providing high-resolution differentiation of isolates [18,24].

Here, we analyze the molecular mechanisms of carbapenem resistance and the essential genes for biofilm formation in *A. baumannii* isolates from the same pulse type and cause of HAIs for over a decade in a big tertiary hospital in Mexico. To provide a useful genotyping framework to make sense of the phenotypic data, we employed the WGS of 76 isolates using an *A. baumannii* cgMLST scheme.

## 2. Materials and Methods

### 2.1. Bacterial Isolates

This descriptive and analytical study was conducted on 76 *A. baumannii* pulse type 22 isolates that were previously obtained from HAIs and were responsible for several outbreaks in patients at the Hospital Civil de Guadalajara (Mexico) from 2007 to 2017. The study received approval from the Ethics and Research Committee of the Facultad de Medicina, Universidad Nacional Autónoma de México (107/2013 and 084/2016), as well as by the Ethics Committee of Hospital Civil de Guadalajara (046/16).

In March 2007, pulse type 22 was identified as sensitive to carbapenems, but in November of the same year, it acquired carbapenem resistance [25]. By 2011, the carbapenem resistance was greater than 70%, a percentage that exceeded 95% in 2015 [26,27]. This collection comprised 63 isolates sampled during 2007–2011 [25,26], 11 isolates sampled in 2016 [27], and 2 isolates sampled in 2017. Out of these 76 isolates, 73 were previously sequenced [28], and the 3 remaining isolates were sequenced for the first time during this study. The DNA was extracted from these three isolates employing a QIAamp mini-kit (Qiagen, Hilden, Germany), following manufacturer’s instructions. The genome sequencing of the DNA from the isolates was conducted on an Illumina MiSeq sequencer with a 2 × 250 bp configuration at the Instituto Nacional de Medicina Genómica (https://www.inmegen.gob.mx/; accessed on 1 May 2020) in Mexico City, Mexico. The first five and last five bases in each read were trimmed using Trim Galore v0.64 (https://github.com/FelixKrueger/TrimGalore; accessed on 6 July 2020). The genomes were assembled with SPAdes v3.13.1 [29] and annotated with Prokka v1.13 [30].

### 2.2. Antimicrobial Susceptibility Profile

The minimum inhibitory concentration (MIC) data of amikacin, gentamicin, cefotaxime, cefepime, levofloxacin, tetracycline, imipenem and meropenem against the 73 isolates used in this study were previously reported [25,27,28] and were used for the analysis of acquired resistance [31]. The MICs of the antimicrobials were determined for the three isolates included in the study for the first time. This was achieved using serial dilution on Mueller–Hinton (MH) agar following the guidelines outlined by the Clinical and Laboratory Standards Institute (CLSI), as previously described [27,32]. The MIC of colistin was determined for the 76 isolates by microdilution broth method using serial two-fold dilutions in a 96-well plate with MH following the guidelines outlined by the European Committee on Antimicrobial Susceptibility Testing (EUCAST) and *Escherichia coli* ATCC 25922 was used as a quality control strain [32,33]. The categories proposed by Magiorakos to characterize the different patterns of resistance were used: multidrug-resistant (MDR), non-susceptibility to at least one agent in three or more antimicrobial categories; extensively drug-resistant (XDR), non-susceptibility to at least one agent in all but two or fewer antimicrobial categories; and pan-drug-resistant (PDR), non-susceptibility to all agents in all antimicrobial categories [31].

### 2.3. Phenotypic Assay to Elucidate the Activity of Efflux Pumps

To investigate whether carbapenem resistance is mediated by efflux pumps, a phenotypic synergy assay was performed for the 76 isolates using meropenem and imipenem and an efflux pump inhibitor, as previously described [34]. Briefly, MH agar plates were prepared with and without the efflux pump inhibitor carbonyl cyanide 3-chlorophenyl-hydrazone (CCCP) at a concentration of 25 µg/mL [25]. The CCCP was dissolved in 1 mL of dimethylsulfoxide (DMSO) for every 2.5 mg of CCCP. Plates were prepared with 9 mL of agar and 1 mL of the carbapenem antimicrobial in two-fold serial dilutions from 0.004 µg/mL to 128 µg/mL. Isolates were adjusted in 2 mL of sterile water to a concentration of 0.5 on the McFarland scale, corresponding to 10^8^ CFU/mL, and diluted 1:10. Using a Steers replicator, 1 µL of the bacterial suspension, corresponding to 10^4^ CFU, was inoculated into the plates and incubated at 37 °C for 24 h. A plate without antimicrobials at the beginning and the end of each series of antimicrobials was used as a control. The positive criteria for phenotypic detection of efflux pump was a 4-fold decrease (or higher) in MIC for carbapenem when CCCP was added [34].

### 2.4. Phenotypic Assay to Determine Biofilm Formation

This assay is based on the ability of bacteria to form biofilms on polyvinylchloride plastic. Biofilm production was assayed in sextuplicate using 96-well plates with Luria Bertani (LB) media, adherent bacteria were stained with crystal violet, and bound dye was quantified by measuring the OD at 595 nm as previously described [35]. Following the recommendations of Stepanović et al. [36], isolates were divided into the following categories: no biofilm producer (0), weak biofilm producer (+ or 1), moderate biofilm producer (++ or 2) or strong biofilm producer (+++ or 3). The results were normalized, taking the negative and positive controls as references. The control optical density (OD_C_) was first determined, which is three standard deviations (SD) above the mean OD of the negative control containing only broth (C−: ODc = mean OD of C− + (3 × SD of C−). Subsequently, the following categories of biofilm production were determined: OD ≤ ODc = no biofilm producer; ODc < OD ≤ 2×ODc = weak biofilm producer; 2×ODc < OD ≤ 4×ODc = moderate biofilm producer; and 4×ODc < OD = high biofilm producer. The ODc determined for the isolates was 0.006315. Biofilm production levels were classified as none = OD ≤ 0.006315; low = 0.006315 < OD ≤ 0.01263; moderate = 0.01263 < OD ≤ 0.025261; and high = 0.025261 < OD. *A. baumannii* ATCC 17961 and ATCC 19606 were used as positive controls, and LB with no inoculum was used as a negative control.

### 2.5. Detection of Antimicrobial Resistance and Biofilm-Formation-Associated Genes and Mutations

The presence and absence of genes related to the production of β-lactamases and virulence factors were assessed for the 76 isolates. The 76 genomes were uploaded to the PubMLST database (https://pubmlst.org/abaumannii/; accessed on 19 January 2022), and gene presence was determined using the BIGSdb Gene Presence tool with the default settings of 70% minimum identity, 50% minimum alignment and 20 BLASTN word size. The mutations associated with antimicrobial resistance and biofilm formation were determined using AMRFinderPlus v 3.10.45 on default settings [37].

### 2.6. MLST, cgMLST and Minimum Spanning Tree

The determination of the STs according to the Oxford and Pasteur MLST schemes was carried out via the PubMLST database (https://pubmlst.org/abaumannii/; accessed on 23 February 2022) [16,17,38]. The PubMLST database was used to assign cgSTs for the cgMLST scheme consisting of 2133 loci. Of note, recently, the cgMLST v1 scheme for *A. baumannii* was created in the PubMLST database based on a modified, reduced set of loci compared to those described in Higgins et al. The minimum spanning tree analysis was generated and visualized using GrapeTree, considering all the loci in the cgMLST v1 scheme as previously described [38,39]. Briefly, the 76 isolates were submitted to PubMLST.org (accessed on 11 April 2022), and the cgMLST v1 scheme was chosen in the GrapeTree plugin. The metadata fields regarding antimicrobial resistance and biofilm formation were exported and overlaid on the generated minimum spanning tree.

## 3. Results

### 3.1. cgMLST Analysis Clusters the 76 Isolates into Seven Clusters from Two International Clones

To achieve a high-resolution genotyping of the isolates, cgMLST analysis was conducted with seventy-three previously sequenced *A. baumannii* isolates [28] plus three newly sequenced isolates, yielding a total of 46 core genome sequence types (cgST). This provided a better resolution than the conventional MLST schemes: the Oxford MLST scheme identified five STs, whereas the Pasteur scheme found two STs. Comparing cgMLST with the Oxford MLST scheme showed that the former has 9.2-fold more resolution than the latter. In order to visualize the clustering of the isolates and their relationship with the international clones (ICs), GrapeTree was used to generate a minimum spanning tree of the 76 isolates using all the loci in the cgMLST scheme. The cgSTs were grouped into clusters at a single-linkage threshold of 50 loci differences. Seven clusters were obtained and were named here as 791 (*n* = 36), 377 (*n* = 30), 792 (*n* = 3), 795 (*n* = 3), 796 (*n* = 2), 793 (*n* = 1) and 794 (*n* = 1) (Figure 1). Note that as these were single-linkage clusters, the cluster names are liable to change over time as groups can merge with the addition of new data. The two larger clusters were 791, with 47.37% (36/76) of the isolates, and 377, with 39.47% (30/76) of the isolates, which mainly but not exclusively included Oxford ST417 in the first case and ST208/1806 in the second case.

The 76 isolates belonged to two important ICs. Cluster 792 corresponded to IC5, which is endemic in Latin America [40], whereas the other six clusters belonged to the global IC2, which is extensively distributed worldwide. When changing the clustering threshold from 50 to 25 alleles, only one additional cluster appeared, where the cluster 377 was split into two groups. This indicated that most of the cgST clusters had a maximum difference of 25 alleles. Increasing the threshold to 100 or 200 alleles merged all the clusters except for 792. This is indicative of a low allelic diversity variance within the clusters belonging to the IC2 but a higher variance between cluster 792, from IC5, and the rest of the IC2 clusters. The higher resolution of cgMLST allowed us to establish the distribution of the 76 isolates, with six related clusters belonging to the IC2 and a seventh more distantly related cluster, which is part of the IC5 (Figure 1).

### 3.2. The Isolates Were Highly Prevalent in Surgery and Intensive Care Units

We analyzed the ward and source of the isolates. They were taken from 10 different sources and 14 different wards. Wound secretions (22.37%; 17/76) and blood cultures (21.05%; 16/76) provided most of the samples. The isolates were scattered throughout the different wards of the hospital, the predominant ward being surgery (30.26%; 23/76) and the intensive care unit (19.74%; 15/76). There was no association between the source or department and cgMLST clusters. In addition, the year in which the isolates were sampled was analyzed for the purpose of better understanding their transmission. The isolates were sampled over 10 years, from 2007 to 2017, specifically between 2007 and 2011 and between 2016 and 2017.

When the isolation years were overlaid on the cgST clusters, isolates clustered according to sampling date. For instance, IC2 isolates sampled in the same year could be found grouped in similar parts of the tree, and as one considers more parts of the tree, the variance in sampling date increases. Interestingly, the three isolates on the IC5 branch were present from 2009, 2011 and 2016. This was the only point on the tree where the 2009 and 2011 isolates were not connected to isolates sampled the same year, differentiating the IC5 from the IC2 (Figure 2).

These findings suggest that isolates were distributed in many wards but were predominantly found in surgery and intensive care units. Furthermore, isolates belonging to IC2 showed an association between genetic similarity and sampling date.

### 3.3. High Antimicrobial Resistance Was Distributed across Clusters

The antimicrobial susceptibility profile was determined for routine antimicrobial therapy for the 76 isolates. Using the antimicrobial panel and the classification scheme described by Magiorakos et al. [31], 43.42% (33/76) of the isolates were classified as PDR, while 56.58% (43/76) were XDR. Notably, none of the isolates were classified as MDR or susceptible. Non-susceptibility phenotypes, which were intermediate and resistant isolates, demonstrated high resistance to meropenem (97.37%; 74/76), imipenem (98.68%; 75/76) and colistin (50.00%; 38/76). Regarding the MIC of these carbapenems, the MIC_50_ and MIC_90_ for meropenem were both greater than 128 µg/mL, while the MIC_90_ and MIC_50_ for imipenem were both greater than 128 µg/mL. Colistin MIC_90_ was 64 µg/mL, and its MIC_50_ was 2 µg/mL.

When the resistance profile was overlaid on the cgMLST minimum spanning tree, both PDR and XDR phenotypes were distributed across the tree except for the IC5 isolates, which were all XDR. In addition, the lower MICs for both imipenem and meropenem were grouped in the same cgST clusters (377 and 795), and susceptible and intermediate isolates were only found in cluster 377 (Appendix A). The PDR and XDR profiles of the isolates were defined by colistin resistance and not carbapenems because most strains were resistant to the latter. All isolates were resistant to both imipenem and meropenem, except for one strain susceptible to imipenem and two to meropenem, both belonging to the cgST cluster 377. Colistin resistance showed a decrease across the first years since all isolates from 2007 were resistant, but from 2008 onwards, resistant and susceptible isolates remained with similar proportions, whereas for the IC5 isolates, two out of the three were susceptible to colistin.

These results show that all the isolates were XDR or PDR, regardless of the cluster they belonged to for IC2, but isolates were only XDR for IC5.

### 3.4. The Oxacillinases Displayed IC-Specific Patterns

The main carbapenem resistance mechanism in *A. baumannii* is the production of β-lactamase enzymes, mainly oxacillinases. All the isolates exhibited the intrinsic *bla*_OXA-51-like_ family gene; specifically, 94.74% (72/76) of the isolates contained the intrinsic *bla*_OXA-66_ (part of the IC2); 3.95% (3/76) possessed the intrinsic *bla*_OXA-65_, with all of them belonging to the IC5; and 1.32% (1/76) of the isolates possessed the intrinsic *bla*_OXA-83_, also within the IC2.

The predominant acquired antimicrobial resistance genes in carbapenem-resistant isolates of *A. baumannii* were the *bla*_OXA-143-like_ family and the *bla*_OXA-72_, which belongs to the *bla*_OXA-24/40-like_ family, both with a 90.79% (69/76); followed by the *bla*_OXA-398_, which belongs to the *bla*_OXA-23-like_ family, with a 9.21% (7/76). These oxacillinases were found in isolates belonging to both IC2 and IC5. The *bla*_OXA-143-like_ family has been described as an efficient penicillinase, oxacillinase, and carbapenemase.

The oxacillinase pattern was specific for each IC (Table 1). When focusing on the IC2, 87.67% (64/73) of the isolates had a combination of the *bla*_OXA-72_ and the intrinsic *bla*_OXA-66_; 5.48% (4/73) had a combination of the *bla*_OXA-398_ and the intrinsic *bla*_OXA-66_; 2.74% (2/73) had a combination of the *bla*_OXA-398_, the *bla*_OXA-72_ and the intrinsic *bla*_OXA-66_; 2.74% (2/73) only had the intrinsic *bla*_OXA-66_; and 1.37% (1/73) only had the intrinsic *bla*_OXA-83_. The IC5 had two different oxacillinase combinations: 66.67% (2/3) of the isolates had a combination of the *bla*_OXA-72_ and the intrinsic *bla*_OXA-65_, and 33.33% (1/3) had a combination of the *bla*_OXA-398_, the *bla*_OXA-72_ and the intrinsic *bla*_OXA-65_.

None of the isolates possessed metallo-β-lactamases.

### 3.5. Two Clusters Have a Higher Carbapenem-Resistance-Associated Efflux Pump Activity

As one of the mechanisms involved in carbapenem resistance is the overexpression of efflux pumps, the activity of efflux pumps was analyzed: 18.42% (14/76) of the isolates were positive for the efflux pump expelling one or both carbapenems, 16% (12/75) for imipenem and 4% (3/75) for meropenem. Moreover, the phenotype of efflux pump activation for one or both carbapenems was always lower than 23% each year examined, except for 2011, when it reached a peak of 58.33% (7/12) for imipenem and 75% (9/12) for meropenem. This was consistent with the 100% resistance rate for carbapenems that year.

When the efflux pump phenotype was overlaid on the cgMLST tree, those isolates with a positive efflux pump were mainly found in cluster 377 (11/12) for imipenem and in cluster 795 (2/3) for meropenem from the IC2. IC5, which consists entirely of cluster 792, just had one positive isolate for imipenem, even though all three isolates were carbapenem-resistant. Taken together, these results suggest that the efflux pump phenotype was associated with these two clusters but not with the resistance profile (Table 2).

The presence of the efflux pump encoding genes in the 76 isolates was examined, and 100% (76/76) of the isolates contained the genes for AdeABC (*adeA*, *adeB*, and *adeC* genes) and its positive regulator AdeRS (*adeR* and *adeS* genes); AdeFGH (*adeF*, *adeG*, and *adeH* genes) and its repressor AdeL (*adeL* gene); and AdeIJK (*adeI*, *adeJ*, and *adeK* genes) and AdeN, its repressor (*adeN*). None of the isolates had the *adeD* and *adeE* genes, which make up the AdeDE efflux pump.

### 3.6. Most of the Isolates Were Strong Biofilm Producers and Contained Biofilm-Associated Genes

Biofilm formation is a principal virulence factor of *A. baumannii*, contributing to the establishment of infection. From the isolates investigated here, 90.79% (69/76) produced strong biofilms, 1.32% (1/76) produced weak biofilms and 7.89% (6/76) produced no biofilm. Each year, more than 87% of the isolates produced a substantial biofilm, reaching 100% in 2010, 2011 and 2017 (1/1, 12/12, and 2/2 isolates, respectively). Less than 13% of all isolates across all years were non-biofilm producers. The poor biofilm producers were all isolated in 2009 (6.25%; 1/16). Only clusters 377 and 791 from IC2 had non-biofilm producers, whereas all the isolates in IC5 were biofilm producers. Therefore, only seven isolates from IC2 clusters 377 and 791 formed low or no biofilms, whereas the rest of the isolates formed substantial biofilms that were likely to be clinically relevant.

There was the presence of biofilm-related genes in the genomes of all isolates (76/76), which encode for four molecules, they were the following: (1) the *bfmR* and *bfmS* genes of the two components system BfmRS; (2) the *csuAB*, *csuA*, *csuB*, *csuC* and *csuD* genes of the *csuA/BABCDE* pili operon; (3) the *pgaA*, *pgaC* and *pgaD* genes of the *pgaABCD* operon; and (4) the *ompA* outer membrane protein gene. Only three isolates had the *csuE* gene, which is part of the *csuA/BABCDE* operon, and only one isolate lacked the *pgaB* gene, which is part of the *pgaABCD* operon. Finally, 47.37% (36/76) of the isolates had the adhesin *bap* gene.

## 4. Discussion

Although conventional MLST schemes are widely employed for genotyping many bacterial pathogens, cgMLST provides a much higher resolution [18,41]. This is critical for *A. baumannii* because the conventional seven-loci MLST schemes present difficulties in genotyping isolates at a clinically relevant resolution [18,22,42]. The Oxford scheme provides most discrimination but fails to accurately group some isolates due to multiple copies of some loci [18,22] and recombination within others [22,42]. The Pasteur MLST scheme has a lower resolution, which makes it difficult to track the actual number of genotypes [18,22]. This study used a cgMLST scheme to characterize 76 carbapenem-resistant *A. baumannii* that caused HAIs and outbreaks over a ten-year period in a Mexican hospital. We identified seven clusters at a single-linkage threshold of 50 loci differences, six belonging to IC2 (cgST 791, 377, 795, 796, 793 and 794) and one to IC5 (cgST 792). Both are internationally distributed clones with high antimicrobial resistance, but IC2 can be found globally, whereas IC5 is predominant in Latin America [5,40,43]. Clusters 791 and 377 were the largest, with more than 80% of the isolates, while cluster 792 had only three isolates. The differences in clusters according to the methods used (PFGE, MLST schemes, and cgMLST) agree with previously published comparisons of PFGE, Pasteur and Oxford MLST and cgMLST [18]. This study focused on one pulse type (same as here) and obtained a higher number of clusters when using the Oxford MLST scheme and even more with the cgMLST. Even though the number of isolates analyzed in that study and the present study was comparable, the former only obtained one ST when using the Pasteur MLST scheme, whereas two were obtained here. Despite the fact that these isolates were from China and Mexico, respectively, the prevalent Pasteur ST2 in the present study, belonging to the IC2, was the same as the one described before, highlighting the importance of this genotype and its global distribution [43]. It should be noted that the present study used a threshold of 50 loci differences for the cgMLST clustering, whereas, in the former study, the threshold used was not reported.

The capability to identify temporal and spatial patterns at high resolution rapidly is one of the main benefits of using cgMLST. Our analysis demonstrates that isolates of IC2 follow a temporal pattern, whereas those of IC5 do not. The cgSTs were widely distributed in terms of ward and source. Despite this, more than a third of the isolates were from wound secretions or blood cultures, and half of the isolates were samples obtained in the surgery or ICU wards. This is consistent with studies that show a higher prevalence of *A. baumannii* in interventional wards such as surgery [44,45]. Additionally, it emphasizes the significance of *A. baumannii* as an opportunistic pathogen because it is more common in patients who are immunocompromised with bacteremia [45].

The antimicrobial resistance profile of the isolates was extremely high, all of them being either XDR or PDR. These findings differ from previous studies, where the proportion of isolates with the XDR (27–30%) and PDR (14–30%) categories were substantially lower [46,47]. Furthermore, most of the isolates were carbapenem-resistant, with only cluster 377 containing susceptible isolates. This is especially concerning, as carbapenems are an antimicrobial of last resort for treating MDR-resistant infections. It is noteworthy all isolates in IC5 cluster 792 were XDR and carbapenem-resistant. We observed that colistin resistance was also very high, with half of the isolates resistant to polymyxin. Colistin resistance is not widely reported, but an increase has been detected in some hospitals, and one European report stated a similar percentage (almost 50%) of colistin resistance as shown in this study [46]. The high resistance phenotype of these isolates poses a complicated scenario, as there are hardly any options to treat infections caused by these highly resistant isolates.

The production of metallo-β-lactamases and oxacillinases is one of the best-studied carbapenem resistance mechanisms in *A. baumannii*, with oxacillinases its main mechanism for carbapenem resistance. Although no MBL genes were found in any of the 76 isolates, OXA genes were identified. This is consistent with the published literature, which indicates that MBLs are less prevalent than OXAs [48]. The intrinsic *bla*_OXA-51-like_ family gene was found in all isolates; specifically, the IC2 isolates mostly possessed the intrinsic *bla*_OXA-66_, with one isolate having the intrinsic *bla*_OXA-83_, whereas the IC5 isolates only exhibited the intrinsic *bla*_OXA-65_. These relationships between the IC2 and the intrinsic *bla*_OXA-66_ and IC5 and the intrinsic *bla*_OXA-65_ have been described previously [40,49].

Notably, the presence of the *bla*_OXA-143-like_ family was identified. This oxacillinase was first reported in Brazil in 2004 and then in Peru in isolates from 2014, but there are no data on this oxacillinase anywhere else until our Mexican isolates from 2007 to 2017 [43,50,51]. The *bla*_OXA-72_, a member of the *bla*_OXA-24/40-like_ family, was the second most frequent oxacillinase observed in this study [25,52,53]. While the *bla*_OXA-23-like_ family is widely disseminated worldwide and has been frequently reported in Mexico, our results show a low prevalence of this oxacillinase [54,55]. Interestingly, the *bla*_OXA-398_ from the *bla*_OXA-23-like_ family was found in seven isolates belonging to both IC2 and IC5. This oxacillinase has been reported in 2023 for the first time in Mexico [56] and in China in 2020 among IC2 isolates [57].

The efflux pump activity was almost exclusively associated with cluster 377, which had a similar distribution of PDR and XDR isolates and was a member of the IC2. The efflux pump activity did not appear to be related to the XDR and PDR resistance profiles of the isolates. There was a discrepancy between the efflux pump activity and genotype, with 18% of the isolates displaying the phenotype and 100% displaying the genotype. This can be explained by the fact that, despite possessing all the genes required for the efflux pumps, which are intrinsic, the isolates also possessed the corresponding negative regulators, preventing their expression. An RT-qPCR experiment could be performed to determine gene expression and its impact on this phenotype.

Almost all the isolates were strong biofilm producers, with only clusters 377 and 791 producing low or no biofilm. Multiple studies have reported considerable variation in biofilm formation among isolates. However, while up to 60% of isolates have shown to exhibit strong biofilm production in previous studies [58,59], our study found that over 90% of isolates were strong biofilm producers, indicating a significant difference. The fact that all the isolates belong to the same pulse type 22 explains the high percentage of biofilm-producing isolates (90%), implying that it was a common trait that has been maintained over time. The deletion of the *csuA/B* and *csuE* genes prevents CsuA/BABCDE pilus assembly, and *csuE* has been found in strong biofilm producer isolates [60,61,62,63]. These reports contradict our findings, which show that the presence of *csuE* had no effect on biofilm formation. This could be due to a negligible participation of *csuE* in pili formation or the compensatory consequence of the presence of other biofilm-related genes. Another gene associated with biofilm formation is *bap*, which was found in approximately half of the strong biofilm producer isolates in this study [61]. The lack of association between *bap* and biofilm formation in almost half of the isolates could also be due to compensatory mechanisms related to other genes, such as *ompA*, which has also been shown to influence biofilm formation [61].

One limitation of this investigation is that all isolates selected in this study share the same pulse type and belong to two closely related STs, potentially influencing the phenotypic and genotypic similarities between them.

## 5. Conclusions

In this report, we utilized sequencing data to dissect the carbapenem resistome and to analyze the biofilm formation capacity of *A. baumannii* isolates of the same pulse type belonging to IC2 and IC5. Genome exploration demonstrated the presence of the *bla*_OXA-143-like_ family, the *bla*_OXA-72_ and the *bla*_OXA-398_ oxacillinase genes in carbapenem-resistant isolates, and a remarkable presence of different biofilm-associated genes and high levels of biofilm formation, both being characteristic factors of pathogenic strains contributing to their endemicity. Importantly, this also represents the first genomic description of the *bla*_OXA-143-like_ family in *A. baumannii* isolates in Mexico.

## Figures and Tables

**Figure 1 microorganisms-11-02316-f001:**
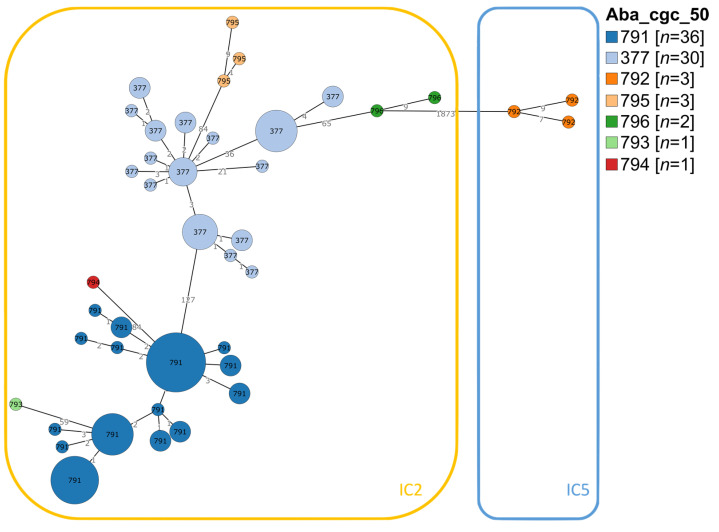
Minimum spanning tree analysis of the 76 isolates from HAIs outbreaks in a tertiary care hospital in Mexico. The tree was constructed using GrapeTree, considering the core genome multilocus sequence type (cgMLST) scheme. The seven clusters named 791, 377, 792, 795, 796, 793 and 794 belong to the two international clones (IC) IC2 and IC5.

**Figure 2 microorganisms-11-02316-f002:**
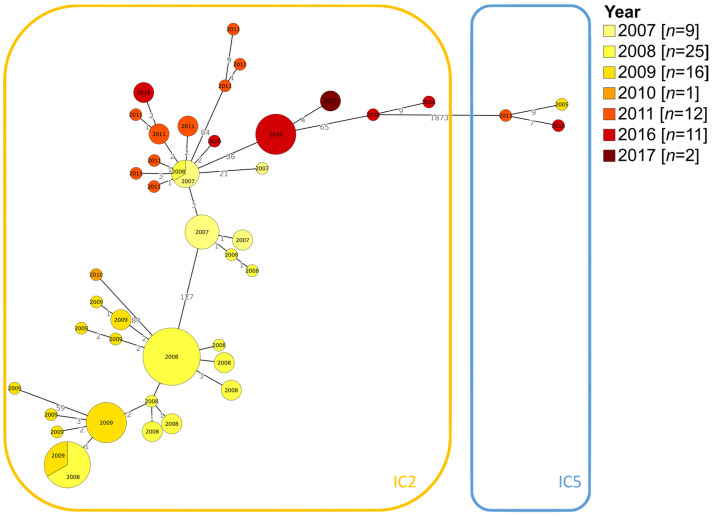
Year of isolation of the 76 samples overlaid on the cgMLST minimum spanning tree. The isolates were sampled from the years 2007 to 2011 and 2016 to 2017 and followed a transitional pattern across the IC2.

**Table 1 microorganisms-11-02316-t001:** Oxacillinase gene presence in the seven cgST clusters. The oxacillinases perform specific patterns among the two ICs.

IC	Oxacillinase Genes	Cluster ^1^	Number of Isolates (*n* = 76)
IC2	*bla*_OXA-72_; *bla*_OXA-66_	791, 793, 794, 377, 796	64
	*bla*_OXA-398_; *bla*_OXA-66_	377, 795	4
	*bla*_OXA-398_; *bla*_OXA-72_; *bla*_OXA-66_	791, 795	2
	*bla* _OXA-66_	377	2
	*bla* _OXA-83_	377	1
IC5	*bla*_OXA-72_; *bla*_OXA-65_ *bla*_OXA-398_; *bla*_OXA-72_; *bla*_OXA-65_	792	2 1

^1^ These are single-linkage clusters; the cluster names are liable to change over time as groups can merge with the addition of new data.

**Table 2 microorganisms-11-02316-t002:** Efflux pump phenotype for imipenem and meropenem in the seven cgST clusters. A positive phenotype was low but mainly found in clusters 377 and 795.

IC	Cluster ^1^	Number of Isolates Positive for Imipenem (*n* = 12)	Number of Isolates Positive for Meropenem (*n* = 3)
IC2	791	0	0
	793	0	0
	794	0	0
	377	11	1
	395	0	2
	396	0	0
IC5	792	1	0

^1^ These are single-linkage clusters; the cluster names are liable to change over time as groups can merge with the addition of new data.

## Data Availability

The authors confirm all supporting data, code and protocols have been provided within the article or through supplementary data files and are available on the PubMLST website. The datasets analyzed for this study can be found in the NCBI under the BioProject numbers PRJNA664412 (https://www.ncbi.nlm.nih.gov/bioproject/PRJNA664412) and PRJNA899659 (https://www.ncbi.nlm.nih.gov/bioproject/PRJNA899659).

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
