# Peer review of "Acinetobacter baumannii IC2 and IC5 Isolates with Co-Existing blaOXA-143-like and blaOXA-72 and Exhibiting Strong Biofilm Formation in a Mexican Hospital"

_microorganisms, 2023, doi:10.3390/microorganisms11092316_

Round 1

Reviewer 1 Report

The manuscript entitled "Acinetobacter baumannii IC2 and IC5 Isolates with Co-existing blaOXA-143-like and blaOXA-72 and Exhibiting Strong Biofilm Formation in a Mexican Hospital" is an excellent contribution to the field of medicinal chemistry and an alarming global health problem – resistance of microorganisms. The authors have collected and presented comprehensive data by analysis of the phenotype and genotype of seventy-six HAIs and outbreak-causing A. baumannii isolates from a Mexican hospital over ten years, with special attention to the carbapenem resistome and biofilm formation. In summary, the paper is well written, the data presented is thorough and the authors' findings are significant contributions to the field. I recommend accepting the paper in its present form.

Author Response

We would like to extend our sincere gratitude for your positive and encouraging feedback on our manuscript. Your recommendation for acceptance of the paper in its present form is both an honor and a validation of our efforts.

Reviewer 2 Report

The authors present interesting data in a well designed study. For further information:

Have the authors data about the four classes of betalactamases (CLASS A, B, C and D)? 

Have the authors data about aminoglycosides resistance mechanisms (aminoglycosides-modifyng anzymes, target site alteration, ...)?

Author Response

Thank you for your comments and suggestions. For this paper we focus on carbapenem resistance because carbapenem-resistant Acinetobacter baumannii is in the WHO critical priority list. Currently, we are conducting a comprehensive study of the resistome and virulome within this collection of 76 isolates, data which we plan to publish in an upcoming article. We have indeed investigated all four classes of beta-lactamases and aminoglycoside resistance genes. We would like to share with you that the vast majority of beta-lactamase genes belong to class D (as reported in this manuscript). Additionally, we have identified blaTEM from class A, blaADC from class C, and no genes from class B (metallo-beta-lactamases). Regarding aminoglycoside resistance genes, we have identified the presence of aminoglycoside transferase genes aph, aac, armA, ant and aadA1.

Reviewer 3 Report

Here are my comments on the manuscript:

1.      Please insert in the Methods the protocol number of ethics committee approval.

2.      Which criteria was used to evaluate the breakpoint of different antibiotics? EUCAST or Clinical and Laboratory Standards Institute?

3.      In the Methods, the type of study is missing. Please add.

4.      The discussion lacks study limitations. Please add.

Author Response

Thank you for your valuable feedback. We have incorporated the following information into the manuscript:

1. This study received approval from the Ethics and Research Committee of the Facultad de Medicina, Universidad Nacional Autónoma de México, with registration numbers 107/2013 and 084/2016, as well as from the Ethics Committee of Hospital Civil de Guadalajara, with registration number 046/16.

2. Antibiotic susceptibility testing foramikacin, gentamicin, cefotaxime, cefepime, levofloxacin, tetracycline, imipenem and meropenem was conducted following the guidelines outlined by the Clinical and Laboratory Standards Institute (CLSI). Colistin susceptibility testing was performed in accordance with the European Committee on Antimicrobial Susceptibility Testing (EUCAST) recommendations.

3. This is a descriptive, analytical study.

4. One important limitation of our study to consider is that all isolates belong to the same pulse type and are associated with two closely related sequence types (STs). This shared genetic background may influence both. 
